health and disease and epidemiology

COVID-19, bibliometrics, citations, productivity, authorship

**Author for correspondence:**
John P. A. Ioannidis
e-mail: jioannid@stanford.edu

# The rapid, massive growth of COVID-19 authors in the scientific literature

John P. A. Ioannidis[1,2], Maia Salholz-Hillel[2],
Kevin W. Boyack[3] and Jeroen Baas[4]

[1]Departments of Medicine, of Epidemiology and Population Health, of Biomedical Data Science, and of Statistics, and Meta-Research Innovation Center at Stanford (METRICS), Stanford University, Stanford, CA, USA
[2]Meta-Research Innovation Center Berlin (METRIC-B), QUEST, Berlin Institute of Health, Berlin, Germany
[3]SciTech Strategies, Inc., Albuquerque, NM, USA
[4]Research Intelligence, Elsevier B.V., Amsterdam, The Netherlands

(iD) JPAI, 0000-0003-3118-6859; JB, 0000-0001-8005-4153

We examined the extent to which the scientific workforce in different fields was engaged in publishing COVID-19-related papers. According to Scopus (data cut, 1 August 2021), 210 183 COVID-19-related publications included 720 801 unique authors, of which 360 005 authors had published at least five full papers in their career and 23 520 authors were at the top 2% of their scientific subfield based on a career-long composite citation indicator. The growth of COVID-19 authors was far more rapid and massive compared with cohorts of authors historically publishing on H1N1, Zika, Ebola, HIV/AIDS and tuberculosis. All 174 scientific subfields had some specialists who had published on COVID-19. In 109 of the 174 subfields of science, at least one in 10 active, influential (top 2% composite citation indicator) authors in the subfield had authored something on COVID-19. Fifty-three hyper-prolific authors had already at least 60 (and up to 227) COVID-19 publications each. Among the 300 authors with the highest composite citation indicator for their COVID-19 publications, most common countries were USA ($n = 67$), China ($n = 52$), UK ($n = 32$) and Italy ($n = 18$). The rapid and massive involvement of the scientific workforce in COVID-19-related work is unprecedented and creates opportunities and challenges. There is evidence for hyper-prolific productivity.

# 1. Introduction

The acute crisis of COVID-19 has challenged the scientific community to generate timely evidence about the new coronavirus and its pandemic. Interest in COVID-19 has spread rapidly and widely

across the scientific literature and among researchers. Such an 'all hands on deck' response of the scientific workforce during a crisis may have been beneficial in generating ideas and evidence expeditiously. However, many authors publishing on COVID-19 may have lacked proper background expertise. The explosive focus on COVID-19 may have caused some inappropriate 'covidization' of research [1,2], and the resulting research, conducted in such haste, may suffer from low validity [3,4].

Here, we aim to understand which scientific areas and which types of scientists have been most mobilized by the pandemic. The growth of the COVID-19 author cohort is contrasted against what happened in the mobilization of the scientific workforce for five other major infectious diseases. We also probe whether there is evidence of hyper-prolific productivity with some scientists rapidly publishing large numbers of papers. Concurrently, we evaluate scientists who have had the highest citation impact for their COVID-19-related work. Finally, we discuss the implications of this rapid 'covidization' of the research enterprise.

# 2. Methods

We used a copy of the Scopus database [5] extracted on 1 August 2021. COVID-19 publications have been specified as those returned by the query: TITLE-ABS-KEY(sars-cov-2 OR 'coronavirus 2' OR 'corona virus 2' OR covid-19 OR {novel coronavirus} OR {novel corona virus} OR 2019-ncov OR covid OR covid19 OR ncovid-19 OR 'coronavirus disease 2019' OR 'corona virus disease 2019' OR corona-19 OR SARS-nCoV OR ncov-2019) AND PUBYEAR > 2018. We further filtered the dataset using the Elsevier International Center for the Study of Research Lab infrastructure to publications indexed (loaded) in Scopus in 2020 or 2021 only, and with the publication year of 2020 or greater. In order to evaluate publication dates by month, we have used the publication month and year where available. When publication month was either not available or exceeded the indexing date, we used the indexing date. This accounts, for example, for cases where an article is published today, but the official journal issue is due later. Our evaluation is targeted at the date at which publications became available to the public rather than official publication dates.

We considered both publications in peer-reviewed venues and preprints. To avoid double counting of the same item (e.g. a work published both in a peer-reviewed journal and as a preprint, or in two preprint servers), we identified and filtered out duplicates by matching against author names and titles. Our search-based approach applies methods used for unstructured reference linking [6] and ranks documents based on the similarity of fields. We used a combination of author names and titles, as many commonly used fields in reference linking, such as journal title, do not apply. The process first identifies multiple potential duplicate matches based on these fields and then validates the best match based on the overlap between words in the title and author names. After identifying top candidate matches for de-duplication, the process applies a validation step based on the overlap between words in the title and author names. We excluded all preprints that link to either a non-preprint item, such as journal articles, or another preprint with an earlier date. The result of this step is the exclusion of 10 703 preprints.

We further focused on the 3 862 276 authors who have at least one Scopus-indexed publication since 2020 and who have also authored in their entire career at least five Scopus-indexed papers classified as articles, reviews or conference papers. This allows the exclusion of authors with limited recent presence in the scientific literature as well as some author IDs that may represent split fragments of the publication record of some more prolific authors.

## 2.1. Field classification

All authors were assigned to their most common field and subfield discipline of their career. We used the Science Metrix classification of science, which is a standard mapping of all science into 21 main fields and 174 subfield disciplines [7,8].

## 2.2. Influential scientists

We also examined how COVID-19 has affected the publication portfolio of researchers whose work has the largest citation impact in the literature. On the one hand, these scientists are already well established and thus may have less need or interest to venture into a new field. On the other hand, these scientists are also more productive and competitive; therefore, they may be faster in moving into a rapidly emerging,

new important frontier. We used the career-long statistics calculated with the Scopus database of 1 August 2021, using the code as provided with the supplemental data recently published for the most-cited authors across science [9–11]. Each author has been assigned to the main field and main subfield based on the largest proportion of publications across fields, and analysis is restricted to the top 2% authors per Science Metrix subfield. We have developed a composite citation indicator [9,10], and accordingly 170 832 scientists can be classified as being in the top 2% of their main subfield discipline based on the citations that their work received through 2020. Of those, 125 869 were active and had published at least one paper also in 2020 or early 2021.

## 2.3. Topics of prominence

In order to visualize the growth and spread of the COVID-19 scientific literature across scientific fields and over time, we used a graphical mapping of scientific fields that has been previously developed [12] and which places the 333 Scopus journal categories sequentially around the perimeter of a circle. There are 27 high-level categories that are placed first and ordered in a manner that emerges naturally from a meta-analysis of the layouts of other science maps created using multiple databases and methods [13]. Each of the 27 categories is assigned a unique colour. The remaining 306 lower level journal categories are then ordered within the corresponding high-level categories using factor analyses based on citation patterns. Each of the 333 journal categories thus has a fixed position on the perimeter of the circle.

The full Scopus citation graph of well over 50 million articles and 1 billion citation links was used to cluster articles into over 90 000 topics using established methods [14]. Each topic is assigned a position within the circle based on the triangulation of the positions of its constituent papers, each of which takes on the positional characteristics of its journal category. Topics are coloured by their dominant journal category and area-sized proportionally based on the number of objects (e.g. papers, authors) being counted for the particular analysis. This circle of science and topic visualization is used in Elsevier's SciVal tool. For the display of authors per topic, we have assigned authors to one topic by taking the topic with the highest proportion of publications per author.

## 2.4. Comparison against other infectious diseases

We performed Scopus searches for terms reflecting four other infectious diseases that have manifested as epidemics in modern history (H1N1, AIDS, Ebola, Zika) and tuberculosis, an epidemic that is ongoing since ancient times and that has probably resulted in the largest cumulative number of deaths over history compared with any other infectious pathogen. One should cautiously interpret comparisons between different infectious diseases considering also the explosive, pandemic nature of COVID-19 and the relative impact of these various disease entities. We used the search terms TITLE-ABS-KEY('swine flu' OR *h1n1*), TITLE-ABS-KEY(*ebola*), TITLE-ABS-KEY(*zika*) and TITLE-ABS(*tuberculosis*). AIDS requires a more thorough search strategy as keywords, and title searches will yield many false positives for the target disease. We collected papers based on the Fingerprint engine concepts [15]: 'Human Immunodeficiency Virus 1', 'HIV Infection', 'AIDS/HIV', 'HIV-1', 'HIV Prevention', 'HIV Testing' and 'Human Immunodeficiency Virus 1 Reverse Transcriptase'. These concepts are based on a unified controlled thesaurus which among other things addresses the disambiguation of homonyms such as 'hearing aids'.

## 2.5. Prolific authors and authors with high citation impact of their COVID-19 publication record

We also mapped the most prolific authors of the published COVID-19 corpus, and the authors whose COVID-19 publications to date had the highest citation impact.

For prolific productivity, we ranked the authors according to decreasing number of COVID-19 published items. We show detailed data on extremely prolific authors with over 60 COVID-19 published items to date. Hyper-prolific publishing reflects a complex phenomenon and may be generated by true productivity and excellence, but also by misconduct (e.g. gift and honorary authorship), and publication of trivialities or 'salami-slicing' where one body of work is cut into multiple 'least publishable units'. We make no effort to probe the key drivers in each hyper-prolific author. This is not feasible for the broad scope and number of papers considered in our study, plus misconduct is extremely difficult to prove. Nevertheless, we dissected among hyper-prolific authors

whether they published also very large numbers of full papers (articles, reviews and conference proceeding papers) or mostly editorializing and other items that are not full papers.

Citation impact was assessed with the previously proposed citation indicator [9–11] that combines information on six indices: total citations, Hirsch h-index, Schreiber hm-index, citations to single-authored papers, citations to first- or single-authored papers, and citations to first-, single- or last-authored papers. This avoids focusing simply on a single traditional metric such as citations, where it is expected that the authors of the earliest highly cited papers would monopolize the top of the list, even if they had published a single paper and they were co-authors among many other authors. Self-citations are excluded from all calculations [10,11]. We present descriptive data on the institution, country and two most common scientific subfields (per Science Metrix classification) for the top 300 authors in that list.

We avoid comparisons based on statistical tests, as the analyses presented here are descriptive and exploratory.

# 3. Results

## 3.1. COVID-19 papers and authors

As of 1 August 2021, Scopus classified 210 863 papers as relevant to COVID-19, which accounts for 3.7% of the 5 728 015 papers across all science published and indexed in Scopus in the period 1 January 2020 until 1 August 2021. The 210 863 published items were classified by Scopus as articles (114 625, 54%), letters (23 029, 11%), reviews (20 641, 10%), preprints from ArXiv, SSRN, BioRxiv, ChemRxiv and medRxiv that could not be matched to other publications (17 953, 9%), notes (12 125, 6%), editorials (10 419, 5%), conference papers (2578, 3%) and other items (4813, 2%).

The 210 863 COVID-19 papers include 720 801 unique authors (with different Scopus IDs), amounting to over 7% of the 9 736 088 author IDs who have published at least 1 paper of any type and on any topic in 2020 or early 2021. The most common countries of these 720 801 authors were USA ($n$ = 143 917), China ($n$ = 72 385), UK ($n$ = 50 392), Italy ($n$ = 45 304), India ($n$ = 34 211) and Spain ($n$ = 29 954) accounting for a total of 376 163 authors (52%). China had more authors involved in COVID-19 papers until May 2020, after which the USA surpassed China.

Among the 3 862 276 authors who have published anything that is Scopus-indexed in 2020 or early 2021 and who have also authored in their entire career at least five Scopus-indexed papers that are classified as articles, reviews or conference papers, by the end of July 2021, 360 005 of these authors (9.3%) had at least one published and indexed COVID-19 paper.

## 3.2. Scientific fields and subfields

Among the 3 862 276 authors, researchers from Public Health and Clinical Medicine (based on their career-long Science Metrix main field) published on COVID-19 at the highest rate: 20.6% (15 886/77 292) of Public Health authors and 17.7% (208 147/1 178 036) of Clinical Medicine authors published COVID-19 research by the end of July 2021. However, publishing COVID-19 research was seen across all 21 major fields. The lowest percentage was seen in the field of Physics & Astronomy (1.7%), from which even 5 364 authors had COVID-19 publications. At the subfield discipline level, the highest COVID-19 publication rate of authors was seen (table 1) in Emergency and Critical Care Medicine (37.00%). However, such rates were higher than 10% (i.e. at least one in 10 authors in that field had published on COVID-19) in 75 subfield disciplines and higher than 5% (i.e. at least one in 20 authors) in 107 subfield disciplines. All 174 subfields had one or more authors publishing on COVID-19. Electronic supplementary material, table S1 gives detailed data for COVID-19 publication rates of authors across all subfield disciplines.

Twenty-eight per cent of the authors published their COVID-19 research primarily in a subfield discipline that was not among the top three subfield disciplines where they had published most commonly during their career. Sometimes the fields of expertise of authors seemed remote from COVID-19, e.g. an expert on solar cells publishing on the epidemiology of COVID-19 in healthcare personnel. Even experts specializing in their past work on remote disciplines such as fisheries, ornithology, entomology or architecture had published on COVID-19.

**Table 1.** Subfields with highest rates of authors publishing on COVID-19.[a]

| subfield | number of authors | authors with COVID-19 paper(s) | % | number of influential authors | influential authors with COVID-19 paper(s) | % |
|---|---|---|---|---|---|---|
| emergency and critical care medicine | 17 450 | 6457 | 37.00 | 523 | 337 | 64.44 |
| anaesthesiology | 18 365 | 6235 | 33.95 | 669 | 264 | 39.46 |
| virology | 31 279 | 10 069 | 32.19 | 1004 | 562 | 55.98 |
| epidemiology | 3830 | 1176 | 30.70 | 143 | 57 | 39.86 |
| applied ethics | 2696 | 819 | 30.38 | 86 | 51 | 59.30 |
| respiratory system | 27 030 | 7744 | 28.65 | 930 | 471 | 50.65 |
| general and internal medicine | 54 981 | 15 639 | 28.44 | 2593 | 1323 | 51.02 |
| allergy | 7147 | 1978 | 27.68 | 254 | 149 | 58.66 |
| medical informatics | 6807 | 1854 | 27.24 | 221 | 113 | 51.13 |
| public health | 31 577 | 8220 | 26.03 | 939 | 434 | 46.22 |
| geriatrics | 5129 | 1301 | 25.37 | 172 | 89 | 51.74 |
| microbiology | 80 988 | 18 518 | 22.87 | 2447 | 854 | 34.90 |
| surgery | 43 798 | 9825 | 22.43 | 1522 | 597 | 39.22 |
| cardiovascular system and hematology | 84 330 | 18 264 | 21.66 | 2989 | 1288 | 43.09 |
| tropical medicine | 15 754 | 3298 | 20.93 | 480 | 210 | 43.75 |

[a]The subfields shown are those with the highest proportions of authors with COVID-19 papers among all authors. See Methods for definition of being an influential author.

## 3.3. Influential scientists and COVID-19 publications

Influential scientists were even more likely to have published COVID-19 research (electronic supplementary material, table S2). Among the 125 869 influential scientists active in publishing in 2020 or early 2021, 23 520 (18.7%) had COVID-19 publications in 2020 or early 2021. The publication rate was the highest in the fields of Public Health (39.7%) and Clinical Medicine (34.4%). Among subfield disciplines, the highest publication rate of such active, influential authors was seen (table 2) in, Emergency & Critical Care Medicine (64.4%), Applied Ethics (60.2%) and Allergy (59.0%). However, publication rates were higher than 10% (i.e. at least one in 10 authors in that field had authored something on COVID-19) in 109 of 174 subfield disciplines across science and higher than 5% (i.e. at least one in 20 authors) in 134 subfield disciplines.

## 3.4. Topics of prominence

Figure 1 shows the growth and spread of COVID-19 papers, authors of COVID-19 papers and high-impact authors of COVID-19 papers (those who belong to the top 2% of impact, as discussed previously) across scientific topics. As shown, there is a strong response of the literature and of the scientific workforce in some specific thematic areas, but there is also increasing and substantial involvement of scientists and respective publications, even in remote topics.

## 3.5. Comparison with other infectious diseases

As shown in figure 2, the massive growth of authors publishing on COVID-19 has been far more rapid and prominent than the growth of the publishing scientific workforce retrieved with the searches for terms reflecting five other infectious diseases. None of the other five infectious diseases managed to attract more than one-tenth the number of new authors in a single year (maximum: HIV/AIDS, 17 968 new authors publishing in 2016 who had not published HIV/AIDS items before) compared with the number of authors recruited by COVID-19 (245 222 authors in 2020) (figure 2*a*). Moreover, none of the other five infectious diseases reached such a large number of active authors publishing in any single year (maxima of 54 409 active authors on HIV/AIDS in 2016, 28 274 active authors on tuberculosis in 2016, 13 983 active authors on H1N1 in 2011, 10 182 active authors on Zika in 2017 and 6855 active authors on Ebola in 2016).

## 3.6. Productivity for COVID-19 publications

A total of 9809 author IDs in Scopus had 10 or more Scopus-indexed published COVID items. Setting thresholds of at least 15, 20, 25, 30, 40, 50 and 60 items, the numbers amounted to 3661, 1674, 932, 539, 220, 113 and 54 separate author IDs, respectively. Figure 3 shows the distribution of COVID publication frequency of authors and in table 2, the 53 authors with 60 or more COVID-19 published items indexed in Scopus (one author had two separate Scopus author ID files which we merged). Of these 53 extremely prolific authors, five were *BMJ* news journalists (including the author with the highest number of published items, Elisabeth Mahase, $n = 227$ published items), two were editors of the *New England Journal of Medicine*, and one was a journalist at *Option/Bio*. Among the remaining 45 scientists, the most common countries were USA ($n = 7$), UK ($n = 6$), Italy ($n = 6$) and India ($n = 5$). When limited to full papers (articles, reviews and conference proceeding papers), there were seven authors who had published 60 or more such full papers and 50 authors had published 40 or more.

## 3.7. Authors with highest citation impact for COVID-19 publications

Electronic supplementary material, table S3 shows the characteristics of COVID-19 authors ranked with the highest citation impact based on the composite citation indicator for their COVID-19-related publications. Among the 300 authors with the highest composite citation indicator scores, 30 were journalists or editors publishing news stories or editorials in their high-impact general medical or science journals. Most common countries for the remaining authors were USA ($n = 67$), China ($n = 52$), UK ($n = 32$), Italy ($n = 18$), Hong Kong ($n = 14$) and India ($n = 12$). Of the 270 scientists excluding journalists/editors, Microbiology was one of their top 2 publishing Science Metrix subfields for 99, followed by General & Internal Medicine ($n = 60$), Virology ($n = 56$) and Immunology ($n = 34$).

**Table 2.** Extremely prolific authors with at least 60 COVID-19 publications indexed in Scopus by 1 August 2021 (not including eight editors/journalists).

| author | institution | country | COVID-19 items | COVID-19 items (non-preprints) | COVID-19 items article/review/conference papers |
|---|---|---|---|---|---|
| Wiwanitkit, Viroj | Dr D. Y. Patil Vidyapeeth Deemed University, Pune | India | 217 | 217 | 9 |
| Lippi, Giuseppe | Università degli Studi di Verona | Italy | 145 | 139 | 74 |
| Rodriguez-Morales, Alfonso J. | Fundación Universitaria Autónoma de las Américas | Colombia | 139 | 139 | 75 |
| Dhama, Kuldeep | Indian Veterinary Research Institute | India | 118 | 118 | 99 |
| Baden, Lindsey R. | Brigham and Women's Hospital | United States | 93 | 88 | 14 |
| Henry, Brandon Michael | Cincinnati Children's Hospital Medical Center | United States | 92 | 87 | 40 |
| Krammer, Florian | Icahn School of Medicine at Mount Sinai | United States | 90 | 66 | 52 |
| Rezaei, Nima | Research Center for Immunodeficiencies | Iran | 87 | 85 | 56 |
| Raoult, Didier | Aix Marseille Université | France | 83 | 82 | 47 |
| Baric, Ralph S. | The University of North Carolina at Chapel Hill | United States | 82 | 61 | 58 |
| Yuen, Kwok Yung | The University of Hong Kong Li Ka Shing Faculty of Medicine | Hong Kong | 80 | 80 | 69 |
| Hasan, Syed Shahzad | University of Huddersfield | United Kingdom | 80 | 80 | 26 |
| To, Kelvin Kai Wang | The University of Hong Kong, State Key Laboratory of Emerging Infectious Diseases | China | 79 | 75 | 66 |
| Kow, Chia Siang | International Medical University | Malaysia | 77 | 77 | 23 |
| Corman, Victor M. | Charité – Universitätsmedizin Berlin | Germany | 77 | 58 | 50 |
| Khunti, Kamlesh | College of Life Sciences | United Kingdom | 77 | 75 | 44 |
| Bragazzi, Nicola Luigi | York University | Canada | 76 | 68 | 60 |
| McKee, Martin | London School of Hygiene and Tropical Medicine | United Kingdom | 76 | 72 | 29 |
| Lechien, Jerome R. | Université de Mons | Belgium | 76 | 74 | 50 |
| Buonsenso, Danilo | Università Cattolica del Sacro Cuore, Campus di Roma | Italy | 74 | 71 | 48 |
| Lu, Hongzhou | Fudan University | China | 73 | 68 | 60 |
| Greninger, Alexander L. | Fred Hutchinson Cancer Research Center | United States | 71 | 59 | 49 |
| Memish, Ziad A. | Alfaisal University | Saudi Arabia | 69 | 68 | 41 |

(Continued.)

**Table 2.** (*Continued.*)

| author | institution | country | COVID-19 items | COVID-19 items (non-preprints) | COVID-19 items article/ review/conference papers |
|---|---|---|---|---|---|
| Eggo, Rosalind M. | London School of Hygiene and Tropical Medicine | United Kingdom | 69 | 55 | 49 |
| Hung, Ivan | The University of Hong Kong Li Ka Shing Faculty of Medicine | Hong Kong | 68 | 66 | 57 |
| Griffiths, Mark D. | Nottingham Trent University | United Kingdom | 67 | 66 | 45 |
| Koopmans, Marion P. G. | Erasmus MC | Netherlands | 66 | 51 | 41 |
| Chan, Jasper Fuk Woo | The University of Hong Kong Li Ka Shing Faculty of Medicine | China | 65 | 61 | 58 |
| Tiwari, Ruchi | College of Veterinary Science India | India | 65 | 65 | 55 |
| Fabbrocini, Gabriella | Università degli Studi di Napoli Federico II | Italy | 63 | 63 | 20 |
| Cowling, Benjamin J. | The University of Hong Kong Li Ka Shing Faculty of Medicine | Hong Kong | 63 | 52 | 40 |
| Saussez, Sven | Université de Mons | Belgium | 63 | 61 | 39 |
| Ohmagari, Norio | National Center for Global Health and Medicine | Japan | 62 | 59 | 46 |
| Joob, Beuy | Private Academic Practice | Thailand | 62 | 62 | 3 |
| Finsterer, Josef | Messerli Institute | Austria | 62 | 62 | 9 |
| Young, Barnaby Edward | Tan Tock Seng Hospital | Singapore | 62 | 56 | 47 |
| Bruno, Raffaele | Università degli Studi di Pavia | Italy | 62 | 57 | 40 |
| Jerome, Keith R. | Fred Hutchinson Cancer Research Center | United States | 62 | 50 | 41 |
| Harky, Amer | Liverpool Heart and Chest Hospital | United Kingdom | 61 | 61 | 40 |
| Netea, Mihai G. | Radboud University Nijmegen Medical Centre | Netherlands | 61 | 46 | 38 |
| Zangrillo, A. | IRCCS San Raffaele Scientific Institute | Italy | 61 | 61 | 41 |
| Plebani, Mario | Azienda Ospedale Università Padova | Italy | 60 | 54 | 33 |
| Alter, Galit | Massachusetts Institute of Technology | United States | 60 | 42 | 37 |
| Vaishya, Raju | Indraprastha Apollo Hospitals | India | 60 | 60 | 37 |
| Lye, David Chien Boon | Tan Tock Seng Hospital | Singapore | 60 | 55 | 48 |

**Figure 1.** Topics of prominence for COVID-19 authors and publications. The columns represent the progress of the spread at three different measuring points: by end of February 2020, end of June 2020, end of October 2020 and end of July 2021. The first row represents the spread of authors of COVID-19 papers. The authors are assigned to their most dominant topic in their career. The data are filtered to include only topics with greater than or equal to five authors assigned. The second row shows similarly the topics of the top 2% authors by field according to a composite citations indicator. Only topics with two or more authors are displayed. The third row displays the spread of COVID-19 publications across topics. The minimum threshold for a topic to be displayed is set to five COVID-19 publications. Of note, the author panels show more dispersed distributions than the publication topic panels, suggesting that several authors are moving out of their main career topics to publish on COVID-19.

# 4. Discussion

More than 700 000 scientists (and counting) have published work related to COVID-19. The most influential scientists across science were even more commonly engaged with COVID-19 research. More than one in six active, influential scientists quickly added or adjusted their publishing portfolio to include COVID-19. More than half of the active, influential scientists in several scientific subfields were involved urgently in COVID-19 work, and every single scientific subfield had some scientists publishing on COVID-19.

The rapid and extensive spread of COVID-19 interests across the map of science was unique compared with other major epidemic infectious diseases. A comparison against five other major epidemic infectious diseases showed that none of them came anywhere close to the explosive nature of the involvement of the scientific workforce in COVID-19-related work. This applied even to HIV/AIDS and tuberculosis that have had a far greater cumulative mortality toll. HIV/AIDS has killed over 35 million people and tuberculosis has killed over 1 billion people to date [16,17].

Our data even underestimate the explosive growth of COVID-19-related work, since some papers are published but not yet indexed. Some of this deficit is captured by preprints (a popular method of disseminating information in the COVID-19 era) [18–20], but the COVID-19 literature is substantially larger than what is indexed in Scopus. The COVID-19 Global Literature on Coronavirus Disease database maintained by the World Health Organization included 318 173 published items as of 31 July 2021 (including 23 673 preprints) (https://search.bvsalud.org/global-literature-on-novel-coronavirus-2019-ncov/). It is possible that authors publishing on COVID-19 may be approaching (or have exceeded) a million as of this writing.

Many authors had published an astonishingly large number of COVID-19 items, and 53 had published 60 or more in such short time. Given delays in indexing, these numbers may underestimate

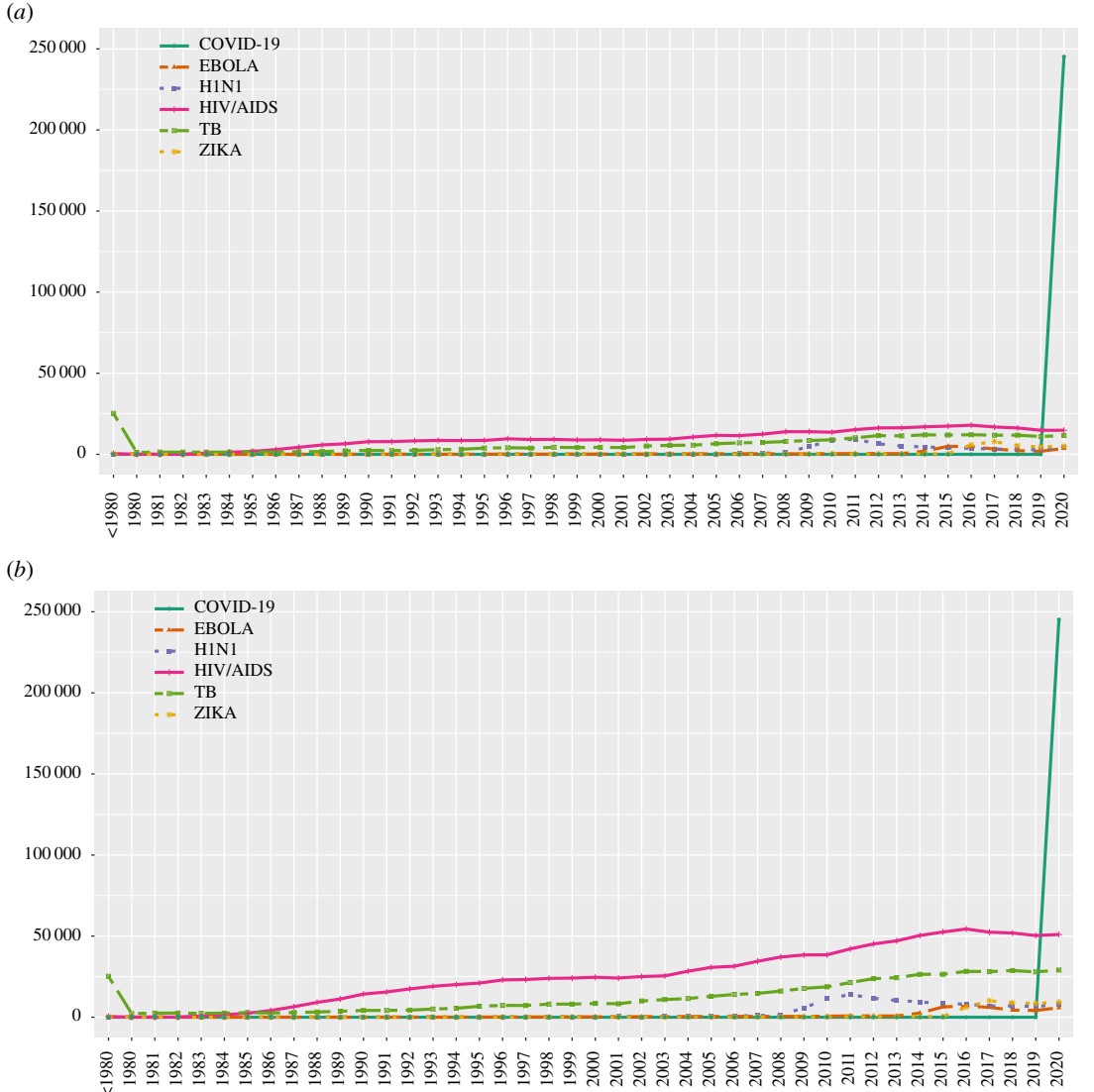

**Figure 2.** Authors publishing on different infectious diseases and COVID-19 every year among the approximately 8 million authors who have published at least five full papers by 2020. (*a*) New author 'cases' per year (authors who publish for the first time on the respective topic, without having any previous publications on this same topic in previous years). (*b*) All active author 'cases' per year (all authors who publish on the respective topic in each year, regardless of whether they have also published on the same topic in previous years or not).

the hyper-prolific productivity. The concentration of hyper-prolific authors in countries like China, Hong Kong and Italy may be related to the early outbreak of the pandemic in these locations, as well as prevalent co-authorship practices in these countries. Some of the unethical and questionable practices surrounding authorship may cluster in specific countries and specific research environments that overtly game and manipulate authorship, through practices such as a gift or honorary authorship. Importantly, meritorious productivity versus sloppiness is difficult to disentangle without examining each case in depth. A large share of the hyper-prolific authors capitalized mostly on copious publishing of editorializing items rather than full papers (articles, reviews or conference proceeding papers).

We also addressed the citation impact of authors for their COVID-19 work. The top ranks included many journalists and editors who published numerous news stories and editorials in their highly visible medical and science journals. This news/editorial function may be helpful. These published items may be readily used for citations, as they are often published well in advance of the scientific work to which they refer. However, the quality, standards and validity of rapidly deployed non-peer-reviewed items are unknown. Flashy news, media and editorializing in both academic journals and the popular press may

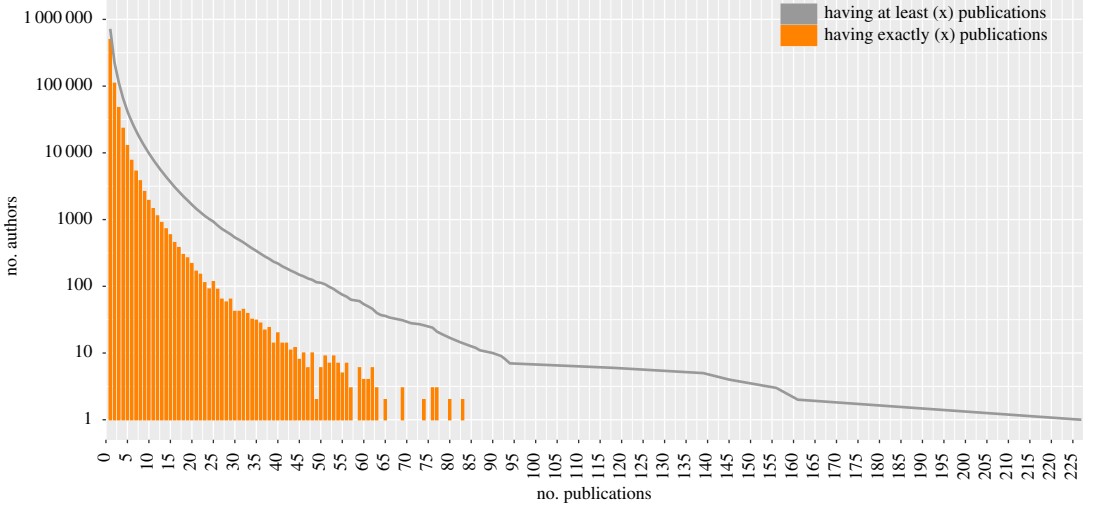

**Figure 3.** Frequency of authors according to the number of COVID-19 publications among the authors in Scopus with five or more publications in total on any topic.

be prominent during the pandemic [21–24]. It is unknown whether non-peer-reviewed news stories and in-house editorials in major journals help safeguard against the 'infodemic' or sometimes contribute to make things worse. Excluding journalists and editors of prestigious journals, the key countries of the authors with the highest composite citation indicator tended to be similar to the countries of the most prolific authors. A few subfields accounted for the lion's share of the authors with the highest composite citation indicator.

The rapid response of the scientific community to crisis is largely a welcome phenomenon. Many scientists quickly focused their attention to an urgent situation and an entirely new pathogen and disease. This demonstrates that the scientific community has sufficient flexibility to shift attention rapidly to major issues. Much was swiftly learned on COVID-19. The quality of the published work was not assessed in our analysis, given the broad scope and huge diversity of the included papers. Nevertheless, many surveys of the quality of COVID-19 publications already exist [15,25–37]. Although existing surveys of the quality of COVID-19 research do not cover all subfields of investigation and quality is often difficult to measure precisely, the consistent finding of the high prevalence of low-quality studies across very different types of study designs suggests that a large portion (perhaps even the large majority) of the immense and rapidly growing COVID-19 literature may be of low quality. Moreover, massive productivity has been described in the pre-COVID era, as affecting researchers across many fields [38] and may be a particular feature for COVID-19 research. Extreme productivity would be worrisome if it sacrifices quality.

The spread of COVID-19 publications in topics and authors traditionally working beyond key relevant disciplines further testifies the great attractiveness of COVID-19 as a field of investigation. The favourable aspect of this expansion is the ability to bring in specialists with expertise in diverse fields, fostering interdisciplinarity. There are situations where experts in seemingly extremely remote fields (e.g. music or second-language acquisition) may indeed be relevant to contribute to the COVID-19 literature. For example, experts almost in any field may be fully justified to publish on how COVID-19 impacted their work. However, we have anecdotally noted that many published contributions represent situations of epistemic trespassing, where scientists try to address COVID-19 health and medical questions, although they come from unrelated fields and probably lack fundamental subject matter expertise. In particular, scientists who work with data of any sort may feel entitled that they can handle, analyse and interpret COVID-19-related data. We do not wish to single out specific scientists, since this may be a very common problem. Furthermore, the exact magnitude of this problem is difficult to fathom, because it is impossible to know details on whether specific scientists may have additional training/expertise on disciplines beyond what they have published on in their careers. However, in the absence of relevant subject matter expertise among the authors' teams, the generated research products may be fundamentally flawed [39]. Such fundamentally flawed research may then even pass peer-review, since the same people populate also the ranks of

peer-reviewers. Flaws go beyond retractions, which account for less than 0.1% of published COVID-19 work [40,41].

Furthermore, there has been a rapid mobilization of funding into COVID-19 research, with some areas, e.g. vaccine development, earmarked for urgent work. According to one analysis, until the end of June 2021, $21.7 trillion have been committed to various activities related to the COVID-19 response (https://www.devex.com/news/interactive-who-s-funding-the-covid-19-response-and-what-are-the-priorities-96833). While the vast majority of these funds are not directly related to research, some of this funding may eventually also support research products and publications. Direct research activities amount to $14 billion, plus there are $173 billion committed to vaccines and treatments and $237 billion committed to health systems. This funding may have worked as an additional attractor of scientists to this rapidly expanding field.

Certain limitations should be discussed. First, current Scopus data have high precision and recall (98.1% and 94.4%, respectively) [5], but some authors may be split into two or more records, and some ID records may include papers from two or more authors. These errors may affect single authors but are unlikely to affect the overall picture obtained in these analyses. Second, field and subfield classification follows a well-known established method, though published items are not precisely categorizable in scientific fields. Third, data on the citation impact of COVID-19 authors are too early to appraise with confidence, and the ranking of specific scientists is highly tenuous and can quickly change with relatively small changes in citation counts. The bigger picture of author characteristics rather than specific names should be the focus of these data. Fourth, since many COVID-19 accepted papers are not yet indexed in Scopus, fields with slower publication and indexing may be relatively under-represented in the analyses. Fifth, we used simple terms that are highly specific for the comparative evaluation of other infectious diseases and some relevant papers and authors working on them may have been missed. However, the difference of these other diseases against the explosive nature of COVID-19 authorships is so stark that it would still be very prominent even if some additional authors working on these diseases could be identified. Sixth, given our study design, we cannot tell whether scientists who shift their attention to COVID-19 are abandoning their prior work, or just working additionally on COVID-19. The pandemic has had direct effects on some types of research, e.g. some investigations were suspended during lockdowns. One would need to have a far longer perspective to examine the long-term impact of potential 'covidization' of research upon other scientific disciplines.

The vast number of COVID-19 papers has posed an extra strain to the already stretched availability of peer-reviewers in journals. Moreover, an unknown percentage of these papers represent work that required research ethics review, adding another layer of reviewing burden. As the pandemic matures, the science of COVID-19 should also mature. Important remaining questions can be raised about the extent and duration of this 'covidization' of research [1,2]. Will scientists continue to flock from different disciplines into COVID-19 research? What consequences might this have for other areas of important investigation and could non-COVID-19 topics be unfairly neglected? Is the response proportional to the magnitude of the crisis? What is the validity and utility of all these publications? Tracking both the pandemic and the scientific response to the pandemic will be useful to make decisions about planning for the growth, reallocation of interest and old-versus-new priorities for science and publishing scientists.

Data accessibility. Data used for this project come from Scopus which is a subscription bibliometric database. Scopus can be contacted through its website and clarifications about the data and accessibility queries can be offered through communication with one of the authors (Jeroen Baas, j.baas@elsevier.com). Data from Scopus have been used in background work on standardized composite citation indicators across the scientific workforce and this work, data and code can be found in https://journals.plos.org/plosbiology/article?id=10.1371/journal.pbio.3000918 and in Mendeley (https://dx.doi.org/10.17632/btchxktzyw). Data include multiple productivity (number of papers) and citation metrics. More extensive data than those reported in the main manuscript are available in the electronic supplementary material, tables. They pertain to the number of authors and number of authors with at least one COVID-19-related publication (electronic supplementary material, table S1); number of influential authors and number of them who had at least one COVID-19-related publication (electronic supplementary material, table S2); and scientists with highest composite citation indicator based on their COVID-19 publications indexed in Scopus as of 1 August 2021 (electronic supplementary material, table S3).

The data are provided in the electronic supplementary material [42].

Authors' contributions. J.P.A.I. had the original idea and wrote the first draft of the paper. J.B. analysed the data. All authors interpreted the data and contributed writing the paper and approved the final version. J.P.A.I. is guarantor.

Competing interests. METRICS has been funded by grants from the Laura and John Arnold Foundation. METRIC-B has been funded by a visiting Einstein fellowship from the Einstein Foundation and Stiftung Charite to J.P.A.I. J.B. is an Elsevier employee, and Elsevier runs Scopus which is the source of the data.

Funding. METRICS has been funded by grants from the Laura and John Arnold Foundation. METRIC-B has been funded by a visiting Einstein fellowship from the Einstein Foundation and Stiftung Charite to J.P.A.I.

Data sharing. All the key data are in the manuscript.

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
