## [Peer Review File · Royal Society Open Science]

Review History

RSOS-210389.R0 (Original submission)

Review form: Reviewer 1

Is the manuscript scientifically sound in its present form?

Yes

Are the interpretations and conclusions justified by the results?

Yes

Is the language acceptable?

Yes

Do you have any ethical concerns with this paper?

No

Have you any concerns about statistical analyses in this paper?

No

Recommendation?

Major revision is needed (please make suggestions in comments)

Comments to the Author(s)

1. "Hyper-prolific publishing" is an interesting term but it needs to be defined in the paper because "productivity" in the TRUE sense is not tested in this research paper. The authors are only looking at counts, not TRUE productivity because some of these hyper-prolific authors are likely participating in honorary authorship or gift authorship and this is unethical (and not true productivity). This should be discussed.
2. Please discuss in your paper the topic of salami slicing because I would imagine that many of these papers are really one study chopped up and published as multiple papers.
3. On Page 14 lines 20-33 you touch on "prevalent co-authorship" but really, the TRUE issue is unethical authorship such as gift or honorary authorship. And we know this is a huge problem in some countries. Please discuss.
4. Methods would be easier to understand if a chart or diagram were also used as a visual to explain.
5. I don't view "news stories" and "editorials" as "scientific literature" and think that by including these in the result, your data is skewed. I would suggest not using them in the data set. By including them I don't think it adds value and in fact, confuses the topic.
6. How many of these articles also had a research ethics committee review stated? This would be a great stat because it would show much much extra burden has been added to research ethics committees during this pandemic.
- 7 On Figure 1, the final column of bubbles on the Right side -- I think this should be Feb 2021 not Feb 2020?
8. Fig 2 -- when you compare COVID to other diseases, it is NOT surprising there is a spike for COVID: COVID is a pandemic -- for example, Ebola was not. COVID impacts everyone no matter their age and sexual activity or sexual preference --- HIV is a different story in that regard. H1N1 never became a pandemic either (I think??) Ebola was very regional. Perhaps discuss these things in your paper as you try to explain the Figure.

Review form: Reviewer 2

Is the manuscript scientifically sound in its present form?

Yes

Are the interpretations and conclusions justified by the results?

Yes

Is the language acceptable?

Yes

Do you have any ethical concerns with this paper?

No

Have you any concerns about statistical analyses in this paper?

No

Recommendation?

Major revision is needed (please make suggestions in comments)

Comments to the Author(s)

This paper quantifies the volume of scholarship that has been published in response to the COVID-19 pandemic. At a very general level, the paper gives a more precise accounting of something that is already perceived at an intuitive level, namely, that a lot of people who work in a wide range of areas have pivoted to produce work of some kind on COVID-19. This quantification is interesting, but in its present form I have some trouble seeing broader value to the current manuscript.

For example, as the author's note, from this kind of high-level quantification we cannot infer anything about the quality of the work that is being produced. So the paper doesn't help us understand how much of this work is critical to understanding and responding to a novel pathogen and how much of it is worthless or harmful. The paper notes that in some cases, individuals from disciplines that "seem remote" from COVID-19 have published on this topic but this strikes me as underdeveloped. COVID-19 is unique in that it has directly affected nearly every aspect of society. I can imagine experts in any discipline where people publish papers (e.g., music, philosophy, archeology, second-language acquisition...) writing something about how COVID-19 has impacted their work, steps they have had to take to adapt to the pandemic, and so on. So when the article calls out an example of someone who works on solar cell technology publishing about COVID-19 in health care one wants to know if it is an article about how COVID has affected the solar cell industry, or solar cells could help hospitals who have to deal with COVID or if this is a case of someone from a far-removed field trying to write about the epidemiology of COVID. To be clear, only this last option seems problematic. If the paper cannot say which category this work falls into, then it should be explicit that such work may simply reflect the extent to which COVID-19 has impacted the life and work of scholars from a wide range of disciplines.

The paper compares the volume of scholarship produced on COVID to the scholarship on other medical conditions and this is also potentially interesting, but I kept wondering what this tells us. For example, if I work in a particular field, it may not take much effort to produce one or more articles at the intersection of my current research and COVID-19. This is a kind of opportunistic publication but it would not reflect a significant shift in my research efforts. Because COVID-19 has affected the lives of nearly every person around the globe in a more direct way than most other diseases or pathogens, it is not surprising that we might see this kind of opportunistic publication. What we want to know, rather, would be if scientists were shifting their research such that they were abandoning their prior work to shift to work on COVID-19. That kind of shift might signal that the system is being inefficient in the allocation of its resources. But from publications alone we can't tell whether the latter is happening or if it is just a higher volume of opportunistic publications. It might be interesting to know if COVID scholarship has come at the expense of scholarship in other areas. But there is a confound here in that the pandemic itself has required that some labs close and some researchers suspend work in other areas. More granularity about dynamics of this kind would be informative. But as it stands, comparing the mortality associated with diseases like HIV and TB to COVID and the volume of work published on each strikes me as potentially misleading since publication volume does not necessarily indicate a mismatch of research effort or that resources are being allocated away from more important avenues of research.

The paper talks about the amount of grant money that has flowed into COVID-19 but that is not quantified or linked to specific papers. Without that, it is difficult to determine the extent to

which this volume reflects this flow of funding or just the fact that the pervasive impacts of COVID-19 provide opportunities for nearly everyone to write something about the pandemic.

At the end of the day I found myself fairly ambivalent toward this paper. It quantifies the amount of research being written about COVID-19 but I'm not sure why we care about that, per se. Rather, we care about that for a range of reasons that this paper doesn't really shed any light on. Are people abandoning valuable research in one area to move into COVID-19 research? We don't know. Are they trying to write outside of their expertise or are they grading low-hanging fruit by writing about how COVID-19 has affected some aspect of their work or the subject matter on which they work? We don't know. How much of this research is high vs low-quality? We don't know. These are the things we care about and the authors seem to realize this. But the paper doesn't shed light on these questions. I was excited to read it, but at the end I felt let down-I was hoping the paper would address all of the questions that are listed as open questions at the end.

Decision letter (RSOS-210389.R0)

Dear Dr Ioannidis

The Editors assigned to your paper RSOS-210389 "The rapid, massive growth of COVID-19 authors in the scientific literature" have now received comments from reviewers and would like you to revise the paper in accordance with the reviewer comments and any comments from the Editors. Please note this decision does not guarantee eventual acceptance.

Please submit your revised manuscript and required files (see below) no later than 21 days from today's (ie 29-Jul-2021) date. Note: the ScholarOne system will 'lock' if submission of the revision is attempted 21 or more days after the deadline. If you do not think you will be able to meet this deadline please contact the editorial office immediately.

on behalf of Dr Isayvani Naicker (Associate Editor) and Nick Pearce (Subject Editor)
openscience@royalsociety.org

Reviewer comments to Author:

Reviewer: 1

Comments to the Author(s)

1. "Hyper-prolific publishing" is an interesting term but it needs to be defined in the paper because "productivity" in the TRUE sense is not tested in this research paper. The authors are only looking at counts, not TRUE productivity because some of these hyper-prolific authors are likely participating in honorary authorship or gift authorship and this is unethical (and not true productivity). This should be discussed.
2. Please discuss in your paper the topic of salami slicing because I would imagine that many of these papers are really one study chopped up and published as multiple papers.
3. On Page 14 lines 20-33 you touch on "prevalent co-authorship" but really, the TRUE issue is unethical authorship such as gift or honorary authorship. And we know this is a huge problem in some countries. Please discuss.
4. Methods would be easier to understand if a chart or diagram were also used as a visual to explain.
5. I don't view "news stories" and "editorials" as "scientific literature" and think that by including these in the result, your data is skewed. I would suggest not using them in the data set. By including them I don't think it adds value and in fact, confuses the topic.
6. How many of these articles also had a research ethics committee review stated? This would be a great stat because it would show much extra burden has been added to research ethics committees during this pandemic.
- 7 On Figure 1, the final column of bubbles on the Right side -- I think this should be Feb 2021 not Feb 2020?
8. Fig 2 -- when you compare COVID to other diseases, it is NOT surprising there is a spike for COVID: COVID is a pandemic -- for example, Ebola was not. COVID impacts everyone no matter their age and sexual activity or sexual preference --- HIV is a different story in that regard. H1N1 never became a pandemic either (I think??) Ebola was very regional. Perhaps discuss these things in your paper as you try to explain the Figure.

Reviewer: 2

Comments to the Author(s)

This paper quantifies the volume of scholarship that has been published in response to the COVID-19 pandemic. At a very general level, the paper gives a more precise accounting of

something that is already perceived at an intuitive level, namely, that a lot of people who work in a wide range of areas have pivoted to produce work of some kind on COVID-19. This quantification is interesting, but in its present form I have some trouble seeing broader value to the current manuscript.

For example, as the author's note, from this kind of high-level quantification we cannot infer anything about the quality of the work that is being produced. So the paper doesn't help us understand how much of this work is critical to understanding and responding to a novel pathogen and how much of it is worthless or harmful. The paper notes that in some cases, individuals from disciplines that "seem remote" from COVID-19 have published on this topic but this strikes me as underdeveloped. COVID-19 is unique in that it has directly affected nearly every aspect of society. I can imagine experts in any discipline where people publish papers (e.g., music, philosophy, archeology, second-language acquisition...) writing something about how COVID-19 has impacted their work, steps they have had to take to adapt to the pandemic, and so on. So when the article calls out an example of someone who works on solar cell technology publishing about COVID-19 in health care one wants to know if it is an article about how COVID has affected the solar cell industry, or solar cells could help hospitals who have to deal with COVID or if this is a case of someone from a far-removed field trying to write about the epidemiology of COVID. To be clear, only this last option seems problematic. If the paper cannot say which category this work falls into, then it should be explicit that such work may simply reflect the extent to which COVID-19 has impacted the life and work of scholars from a wide range of disciplines.

The paper compares the volume of scholarship produced on COVID to the scholarship on other medical conditions and this is also potentially interesting, but I kept wondering what this tells us. For example, if I work in a particular field, it may not take much effort to produce one or more articles at the intersection of my current research and COVID-19. This is a kind of opportunistic publication but it would not reflect a significant shift in my research efforts. Because COVID-19 has affected the lives of nearly every person around the globe in a more direct way than most other diseases or pathogens, it is not surprising that we might see this kind of opportunistic publication. What we want to know, rather, would be if scientists were shifting their research such that they were abandoning their prior work to shift to work on COVID-19. That kind of shift might signal that the system is being inefficient in the allocation of its resources. But from publications alone we can't tell whether the latter is happening or if it is just a higher volume of opportunistic publications. It might be interesting to know if COVID scholarship has come at the expense of scholarship in other areas. But there is a confound here in that the pandemic itself has required that some labs close and some researchers suspend work in other areas. More granularity about dynamics of this kind would be informative. But as it stands, comparing the mortality associated with diseases like HIV and TB to COVID and the volume of work published on each strikes me as potentially misleading since publication volume does not necessarily indicate a mismatch of research effort or that resources are being allocated away from more important avenues of research.

The paper talks about the amount of grant money that has flowed into COVID-19 but that is not quantified or linked to specific papers. Without that, it is difficult to determine the extent to which this volume reflects this flow of funding or just the fact that the pervasive impacts of COVID-19 provide opportunities for nearly everyone to write something about the pandemic.

At the end of the day I found myself fairly ambivalent toward this paper. It quantifies the amount of research being written about COVID-19 but I'm not sure why we care about that, per se. Rather, we care about that for a range of reasons that this paper doesn't really shed any light on. Are people abandoning valuable research in one area to move into COVID-19 research? We don't know. Are they trying to write outside of their expertise or are they grading low-hanging

fruit by writing about how COVID-19 has affected some aspect of their work or the subject matter on which they work? We don't know. How much of this research is high vs low-quality? We don't know. These are the things we care about and the authors seem to realize this. But the paper doesn't shed light on these questions. I was excited to read it, but at the end I felt let down- I was hoping the paper would address all of the questions that are listed as open questions at the end.

===PREPARING YOUR MANUSCRIPT===

===PREPARING YOUR REVISION IN SCHOLARONE===

<https://royalsociety.org/journals/authors/author-guidelines/#supplementary-material> to include a suitable title and informative caption. An example of appropriate titling and captioning may be found at https://figshare.com/articles/Table_S2_from_Is_there_a_trade-off_between_peak_performance_and_performance_breadth_across_temperatures_for_aerobic_sc_ope_in_teleost_fishes_/3843624.

Author's Response to Decision Letter for (RSOS-210389.R0)

See Appendix A.

Decision letter (RSOS-210389.R1)

Dear Dr Ioannidis,

It is a pleasure to accept your manuscript entitled "The rapid, massive growth of COVID-19 authors in the scientific literature" in its current form for publication in Royal Society Open Science.

COVID-19 rapid publication process:

We are taking steps to expedite the publication of research relevant to the pandemic. If you wish, you can opt to have your paper published as soon as it is ready, rather than waiting for it to be published the scheduled Wednesday.

This means your paper will not be included in the weekly media round-up which the Society sends to journalists ahead of publication. However, it will still appear in the COVID-19 Publishing Collection which journalists will be directed to each week (<https://royalsocietypublishing.org/topic/special-collections/novel-coronavirus-outbreak>).

If you wish to have your paper considered for immediate publication, or to discuss further, please notify openscience_proofs@royalsociety.org and press@royalsociety.org when you respond to this email.

on behalf of Dr Isayvani Naicker (Associate Editor) and Nick Pearce (Subject Editor)
openscience@royalsociety.org

Appendix A

August 20, 2021

The Editors

Re: RSOS-210389 "The rapid, massive growth of COVID-19 authors in the scientific literature"

Dear Dr. Naicker and Dr. Pearce

We were pleased to hear that Royal Society Open Science is interested in a revised version of our work. We are grateful for the constructive comments of the reviewers. We have addressed all of their suggestions in the current revised version. In more detail:

Reviewer: 1

Comments to the Author(s)

1. "Hyper-prolific publishing" is an interesting term but it needs to be defined in the paper because "productivity" in the TRUE sense is not tested in this research paper. The authors are only looking at counts, not TRUE productivity because some of these hyper-prolific authors are likely participating in honorary authorship or gift authorship and this is unethical (and not true productivity). This should be discussed.

Reply: We fully agree. We have clarified that "Hyper-prolific publishing reflects a complex phenomenon and may be generated by true productivity and excellence, but also by misconduct (e.g., gift and honorary authorship), and publication of trivialities or "salami-slicing" where one body of work is cut into multiple "least publishable units." We make no effort to probe the key drivers in each hyper-prolific author. This is not feasible for the broad scope and number of papers considered in our study, plus misconduct is extremely difficult to prove. Nevertheless, we dissected among hyper-prolific authors, whether they published also very large numbers of full papers (articles, reviews, conference proceeding papers) or mostly editorializing and other items that are not full papers."

2. Please discuss in your paper the topic of salami slicing because I would imagine that many of these papers are really one study chopped up and published as multiple papers.

Reply: agreed. Please see what was added above, including the salami-slicing pattern.

3. On Page 14 lines 20-33 you touch on "prevalent co-authorship" but really, the TRUE issue is unethical authorship such as gift or honorary authorship. And we know this is a huge problem in some countries. Please discuss.

Reply: agreed, as above. We have also elaborated that "Some of the unethical and questionable practices surrounding authorship may also cluster in specific countries and specific research

environments that overtly game and manipulate authorship, through practices such as gift or honorary authorship.”

4. Methods would be easier to understand if a chart or diagram were also used as a visual to explain.

Reply: We have struggled on how this might work, but could not come up with some diagram/visual that would convey enough information without a lot of wordy explanations.

5. I don't view "news stories" and "editorials" as "scientific literature" and think that by including these in the result, your data is skewed. I would suggest not using them in the data set. By including them I don't think it adds value and in fact, confuses the topic.

Reply: As suggested, in Table 2, we have added a new column that shows for each of the hyper-prolific authors the number of full papers on COVID-19 (including only articles, reviews, and conference proceeding papers). We explain this in the Methods section, we comment on the findings in the Results and have added in the Discussion that “A large share of the hyper-prolific authors capitalized mostly on copious publishing of editorializing items rather than full papers (articles, reviews or conference proceeding papers).”

6. How many of these articles also had a research ethics committee review stated? This would be a great stat because it would show much much extra burden has been added to research ethics committees during this pandemic.

Reply: This figure is difficult to extract, because research ethics committee review is very rarely reported, but this does not mean that non-reporting means that no ethical review happened. We have added in the Discussion that “the vast number of COVID-19 papers have posed an extra strain to the already stretched availability of peer-reviewers in journals. Moreover, an unknown percentage of these papers represent work that required research ethics review, adding another layer of reviewing burden.”

7 On Figure 1, the final column of bubbles on the Right side -- I think this should be Feb 2021 not Feb 2020?

Reply: thank you for noticing this, it has been corrected now.

8. Fig 2 -- when you compare COVID to other diseases, it is NOT surprising there is a spike for COVID: COVID is a pandemic -- for example, Ebola was not. COVID impacts everyone no matter their age and sexual activity or sexual preference --- HIV is a different story in that regard. H1N1 never became a pandemic either (I think??) Ebola was very regional. Perhaps discuss these things in your paper as you try to explain the Figure.

Reply: We have added “One should cautiously interpret comparisons between different infectious diseases considering also for the explosive, pandemic nature of COVID-19 and the relative impact of these various disease entities.”

Reviewer: 2

Comments to the Author(s)

This paper quantifies the volume of scholarship that has been published in response to the COVID-19 pandemic. At a very general level, the paper gives a more precise accounting of something that is

already perceived at an intuitive level, namely, that a lot of people who work in a wide range of areas have pivoted to produce work of some kind on COVID-19. This quantification is interesting, but in its present form I have some trouble seeing broader value to the current manuscript.

For example, as the author's note, from this kind of high-level quantification we cannot infer anything about the quality of the work that is being produced. So the paper doesn't help us understand how much of this work is critical to understanding and responding to a novel pathogen and how much of it is worthless or harmful. The paper notes that in some cases, individuals from disciplines that "seem remote" from COVID-19 have published on this topic but this strikes me as underdeveloped. COVID-19 is unique in that it has directly affected nearly every aspect of society. I can imagine experts in any discipline where people publish papers (e.g., music, philosophy, archeology, second-language acquisition...) writing something about how COVID-19 has impacted their work, steps they have had to take to adapt to the pandemic, and so on. So when the article calls out an example of someone who works on solar cell technology publishing about COVID-19 in health care one wants to know if it is an article about how COVID has affected the solar cell industry, or solar cells could help hospitals who have to deal with COVID or if this is a case of someone from a far-removed field trying to write about the epidemiology of COVID. To be clear, only this last option seems problematic. If the paper cannot say which category this work falls into, then it should be explicit that such work may simply reflect the extent to which COVID-19 has impacted the life and work of scholars from a wide range of disciplines.

Reply: we greatly appreciate this criticism and we are fully aligned on these issues with the reviewer. The solar cell example belongs to what the reviewer would call problematic, and examples of such problematic situations are numerous, but we would like to avoid pointing out specific individuals by name since this might be seen as a personal attack, plus it would be unfair to single out one or a few scientists when so many thousands are doing this. We have however clarified the problematic nature of the solar cell example: "Sometimes the fields of expertise of authors seemed remote from COVID-19, e.g., an expert on solar cells publishing on the epidemiology of COVID-19 in healthcare personnel." We have also added in the discussion a clarification: "There are situations where experts in seemingly extremely remote fields (e.g., music or second-language acquisition) may indeed be relevant to contribute to the COVID-19 literature. For example, experts almost in any field may be fully justified to publish on how COVID-19 impacted their work. However, we have anecdotally noted that many published contributions represent situations of epistemic trespassing, where scientists try to address COVID-19 health and medical questions, although they come from unrelated fields and lack fundamental subject-matter expertise. In particular, scientists who work with data of any sort, may feel entitled that they can handle, analyze, and interpret COVID-19-related data. We do not wish to single out specific scientists, since this may be a very common problem. Furthermore, the exact magnitude of this problem is difficult to fathom, because it is impossible to know details on whether specific scientists may have additional training/expertise on disciplines beyond what they have published on in their careers. However, in the absence of relevant subject-matter expertise among the authors' teams, the generated research products may be fundamentally flawed. Such fundamentally flawed research may then even pass peer-review, since the same people populate also the ranks of peer-reviewers."

The paper compares the volume of scholarship produced on COVID to the scholarship on other medical conditions and this is also potentially interesting, but I kept wondering what this tells us. For example, if I work in a particular field, it may not take much effort to produce one or more articles at the intersection of my current research and COVID-19. This is a kind of opportunistic publication but it

would not reflect a significant shift in my research efforts. Because COVID-19 has affected the lives of nearly every person around the globe in a more direct way than most other diseases or pathogens, it is not surprising that we might see this kind of opportunistic publication. What we want to know, rather, would be if scientists were shifting their research such that they were abandoning their prior work to shift to work on COVID-19. That kind of shift might signal that the system is being inefficient in the allocation of its resources. But from publications alone we can't tell whether the latter is happening or if it is just a higher volume of opportunistic publications. It might be interesting to know if COVID scholarship has come at the expense of scholarship in other areas. But there is a confound here in that the pandemic itself has required that some labs close and some researchers suspend work in other areas. More granularity about dynamics of this kind would be informative. But as it stands, comparing the mortality associated with diseases like HIV and TB to COVID and the volume of work published on each strikes me as potentially misleading since publication volume does not necessarily indicate a mismatch of research effort or that resources are being allocated away from more important avenues of research.

Reply: These are important issues and worth discussing, although they go beyond the current work. We have clarified: "Given our study design, we cannot tell whether scientists who shift their attention to COVID-19 are abandoning their prior work, or just working additionally on COVID-19. The pandemic has had direct effects on some types of research, e.g., some investigations were suspended during lockdowns. One would need to have a far longer perspective to examine the long-term impact of potential "covidization" of research upon other scientific disciplines." Regarding the comparison to other infectious diseases, see also the response to the comments of reviewer 1.

The paper talks about the amount of grant money that has flowed into COVID-19 but that is not quantified or linked to specific papers. Without that, it is difficult to determine the extent to which this volume reflects this flow of funding or just the fact that the pervasive impacts of COVID-19 provide opportunities for nearly everyone to write something about the pandemic.

Reply: We have added that "According to one analysis, until the end of June 2021, \$21.7 trillion have been committed to various activities related to the COVID-19 response (<https://www.devex.com/news/interactive-who-s-funding-the-covid-19-response-and-what-are-the-priorities-96833>). While the vast majority of these funds are not directly related to research, some of this funding may eventually also support research products and publications. Direct research activities amount to \$14 billion, plus there are \$173 billion committed to vaccines and treatments and \$237 billion committed to health systems."

At the end of the day I found myself fairly ambivalent toward this paper. It quantifies the amount of research being written about COVID-19 but I'm not sure why we care about that, per se. Rather, we care about that for a range of reasons that this paper doesn't really shed any light on. Are people abandoning valuable research in one area to move into COVID-19 research? We don't know. Are they trying to write outside of their expertise or are they grading low-hanging fruit by writing about how COVID-19 has affected some aspect of their work or the subject matter on which they work? We don't know. How much of this research is high vs low-quality? We don't know. These are the things we care about and the authors seem to realize this. But the paper doesn't shed light on these questions. I was excited to read it, but at the end I felt let down--I was hoping the paper would address all of the questions that are listed as open questions at the end.

Reply: As above, the revision addresses these issues, putting things into perspective. As for quality, we have added several additional references that already show that the quality of much COVID-19 literature is low, and have clarified: "The quality of the published work was not assessed in our analysis, given the broad scope and huge diversity of the included papers. Many surveys of the quality of COVID-19 publications already exist. Although existing surveys of the quality of COVID-19 research do not cover all subfields of investigation and quality is often difficult to measure precisely, the consistent finding of high prevalence of low quality studies across very different types of study designs suggests that a large portion (perhaps even the large majority) of the immense and rapidly growing COVID-19 literature may be of low quality." References added here are the following:

1: Khatter A, Naughton M, Dambha-Miller H, Redmond P. Is rapid scientific publication also high quality? Bibliometric analysis of highly disseminated COVID-19 research papers. *Learn Publ.* 2021 Jun 1;10.1002/leap.1403.

2: Wang D, Chen L, Wang L, Hua F, Li J, Li Y, Zhang Y, Fan H, Li W, Clarke M. Abstracts for reports of randomised trials of COVID-19 interventions had low quality and high spin. *J Clin Epidemiol.* 2021 Jul 2;S0895-4356(21)00205-5.

3: Abbott R, Bethel A, Rogers M, Whear R, Orr N, Shaw L, Stein K, Thompson Coon J. Characteristics, quality and volume of the first 5 months of the COVID-19 evidence synthesis infodemic: a meta-research study. *BMJ Evid Based Med.* 2021 Jun 3;bmjebm-2021-111710.

4: Li Q, Zhou Q, Xun Y, Liu H, Shi Q, Wang Z, Zhao S, Liu X, Liu E, Fu Z, Chen Y, Luo Z. Quality and consistency of clinical practice guidelines for treating children with COVID-19. *Ann Transl Med.* 2021 Apr;9(8):633.

5: Suchá D, van Hamersvelt RW, van den Hoven AF, de Jong PA, Verkooijen HM. Suboptimal Quality and High Risk of Bias in Diagnostic Test Accuracy Studies at Chest Radiography and CT in the Acute Setting of the COVID-19 Pandemic: A Systematic Review. *Radiol Cardiothorac Imaging.* 2020 Jul 30;2(4):e200342.

6: Kuang Z, Li X, Cai J, Chen Y, Qiu X, Ni X; Evidence-based Traditional and Integrative Medicine Working Group for Public Health Emergency. Calling for improved quality in the registration of traditional Chinese medicine during the public health emergency: a survey of trial registries for COVID-19, H1N1, and SARS. *Trials.* 2021 Mar 5;22(1):188.

7: Li Y, Cao L, Zhang Z, Hou L, Qin Y, Hui X, Li J, Zhao H, Cui G, Cui X, Li R, Lin Q, Li X, Yang K. Reporting and methodological quality of COVID-19 systematic reviews needs to be improved: an evidence mapping. *J Clin Epidemiol.* 2021 Feb 28;135:17-28.

8: Quinn TJ, Burton JK, Carter B, Cooper N, Dwan K, Field R, Freeman SC, Geue C, Hsieh PH, McGill K, Nevill CR, Rana D, Sutton A, Rowan MT, Xin Y. Following the science? Comparison of methodological and reporting quality of covid-19 and other research from the first wave of the pandemic. *BMC Med.* 2021 Feb 23;19(1):46.

9: Luo X, Liu Y, Ren M, Zhang X, Janne E, Lv M, Wang Q, Song Y, Mathew JL, Ahn HS, Lee MS, Chen Y. Consistency of recommendations and methodological quality of guidelines for the diagnosis and treatment of COVID-19. *J Evid Based Med.* 2021 Feb;14(1):40-55.

10: Yang S, Li A, Eshaghpour A, Ivanisevic S, Salopek A, Eikelboom J, Crowther M. Quality of early evidence on the pathogenesis, diagnosis, prognosis and treatment of COVID-19. *BMJ Evid Based Med*. 2020 Sep 30;bmjebm-2020-111499.

11: Nieto I, Navas JF, Vázquez C. The quality of research on mental health related to the COVID-19 pandemic: A note of caution after a systematic review. *Brain Behav Immun Health*. 2020 Aug;7:100123.

12: Alexander PE, Debono VB, Mammen MJ, Iorio A, Aryal K, Deng D, Brocard E, Alhazzani W. COVID-19 coronavirus research has overall low methodological quality thus far: case in point for chloroquine/hydroxychloroquine. *J Clin Epidemiol*. 2020 Jul;123:120-126.

Other comments:

Reply: We have added a 100 words summary for this purpose.

Finally, given that almost 5 months lapsed for the peer-review of the paper, we have updated all the numbers/tables/figures with the most up-to-date version of Scopus going up to August 1, 2021. All the key findings and messages of the paper remain unaltered and strengthened, if anything.

We thank you again for the extremely helpful feedback on our original submission and for the opportunity to improve our work. Having made these revisions, we hope that our manuscript would be acceptable for publication in the current version.

Sincerely,

John Ioannidis